# Position: Modular Safety Guardrails Are Necessary for Foundation-Model-Enabled Robots in the Real World

**Joonkyung Kim** [1†] **Wenxi Chen** [2†] **Davood Soleymanzadeh** [1†] **Yi Ding** [2‡] **Xiangbo Gao** [1‡] **Zhengzhong Tu** [1]
**Ruqi Zhang** [2] **Fan Fei** [3] **Sushant Veer** [4] **Yiwei Lyu** [1*] **Minghui Zheng** [1*] **Yan Gu** [2*]

## Abstract

The integration of foundation models (FMs) into robotics has accelerated real-world deployment, while introducing new safety challenges arising from open-ended semantic reasoning and embodied physical action. These challenges require safety notions beyond physical constraint satisfaction. In this position paper, we characterize FM-enabled robot safety along three dimensions: action safety (physical feasibility and constraint compliance), decision safety (semantic and contextual appropriateness), and human-centered safety (conformance to human intent, norms, and expectations). We argue that existing approaches, including static verification, monolithic controllers, and end-to-end learned policies, are insufficient in settings where tasks, environments, and human expectations are open-ended, long-tailed, and subject to adaptation over time. To address this gap, we propose modular safety guardrails, consisting of monitoring (evaluation) and intervention layers, as an architectural foundation for comprehensive safety across the autonomy stack. Beyond modularity, we highlight possible cross-layer co-design opportunities through representation alignment and conservatism allocation to enable faster, less conservative, and more effective safety enforcement. We call on the community to explore richer guardrail modules and principled co-design strategies to advance safe real-world physical AI deployment.

† Equal co-lead contribution. ‡ Equal second-author contribution. * Equal senior advising. This work is not associated with Amazon. [1]Texas A&M University [2]Purdue University [3]Amazon [4]NVIDIA. Correspondence to: Yiwei Lyu <yiweilyu@tamu.edu>, Minghui Zheng <mhzheng@tamu.edu>, Yan Gu <yangu@purdue.edu>.

*Proceedings of the 43$^{rd}$ International Conference on Machine Learning*, Seoul, South Korea. PMLR 306, 2026. Copyright 2026 by the author(s).

## 1. Introduction

Foundation models (FMs), large-scale networks pretrained for broad generalization, are rapidly becoming core components of modern robotic autonomy stacks (Firoozi et al., 2025; Siciliano et al., 2008). They are increasingly used for perception (Gadre et al., 2023), task planning (Zitkovich et al., 2023; Driess et al., 2023), and end-to-end visuomotor control (Kim et al., 2024; Black et al., 2025), enabling open-world semantic reasoning and cross-task generalization that push robots beyond controlled laboratory settings.

Embodiment fundamentally reshapes the safety problem (Kojima et al., 2025; Wu et al., 2024; Grislain et al., 2025). Classical robotics safety often assumes fixed and predefinable constraints (e.g., geometric collision bounds), whereas FM-enabled robots operate in open-ended environments where hazards are context-dependent and specifications evolve (Santos et al., 2025; Liu & Feng, 2024). Risk is further compounded by environmental uncertainty and FM stochasticity (Hafez et al., 2025; Dalrymple et al., 2024).

Moreover, interactions with humans impose safety requirements beyond physical feasibility, including semantic appropriateness, intent alignment, and adherence to social norms (Dragan et al., 2013; Tian & Oviatt, 2021; Brunke et al., 2025). These requirements induce diverse failure modes that cannot be addressed by a single mechanism (Liu et al., 2023): physical safety filters cannot infer that a "knife handoff" is contextually dangerous (Brunke et al., 2025), while semantic reasoners lack real-time enforcement needed to prevent collisions (Kojima et al., 2025). No monolithic safety mechanism reliably addresses all such failures.

A natural response is to learn a single end-to-end safety guardrail that jointly encodes physical, semantic, and intent constraints, but such monolithic solutions remain fragile in practice (Dawson et al., 2023). They are vulnerable to distribution shift as tasks, environments, and safety requirements evolve (Farid et al., 2022; Buysse et al., 2025), often necessitating additional frequently updated or externally imposed safety components (Ren et al., 2023; Peng et al., 2025). Moreover, real-world datasets rarely contain catastrophic safety failures, leaving the most critical modes underrepre-

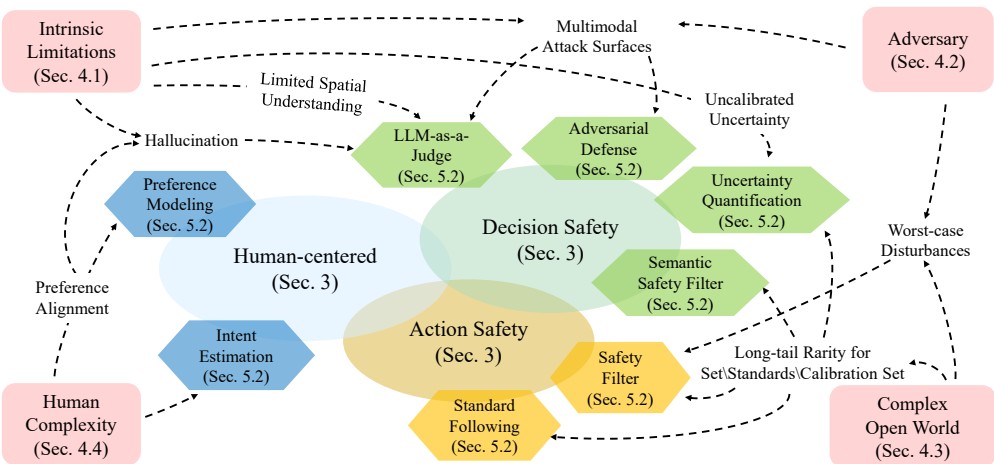

*Figure 1.* Overview of the safety definitions (Sec. 3), source of safety challenges (Sec. 4), and existing alternative methods (Sec. 5).

sented during training (Tölle et al., 2025). These limitations fundamentally constrain the robustness and longevity of purely end-to-end safety solutions.

**In this position paper, we argue that architectural modularity is necessary for the safe and reliable deployment of FM-enabled robotic systems.**

> **Core Claim**
>
> *Under open-world deployment with long-tailed hazards and non-stationary human interaction, safety mechanisms that are either fully embedded within the autonomy model or confined to a single layer of the autonomy stack are structurally insufficient to ensure action, decision, and human-centered safety simultaneously.*

In particular, our *external modularity* separates safety enforcement from FMs to prevent FM uncertainty from affecting safety, while *internal modularity* decomposes safety into specialized mechanisms targeting distinct failure modes.

We ground this claim with a three-dimensional taxonomy of FM-enabled robotics safety: action safety (physical feasibility and constraint compliance), decision safety (semantic and contextual appropriateness), and human-centered safety (conformance to human intent, norms, and expectations), as illustrated in Fig. 1. We argue that non-modular approaches are fundamentally brittle under real-world conditions, where safety requirements drift, are hard to specify a priori, and failures are rare but high impact.

To address these challenges, we propose a two-layer modular safety guardrail architecture (Fig. 2) with (i) a Monitoring and Evaluation Layer that assesses risk across the autonomy stack and (ii) an Intervention Layer that enforces safety through decision-level gating and action-level filtering. This modular design enables principled cross-layer co-design, such as representation alignment and conservatism allocation, allowing more precise and less conservative safety

enforcement, while supporting independent verification, updateability, composability of heterogeneous mechanisms, and systematic coverage across all three safety dimensions.

This paper first reviews the integration of FM into robotic systems (Sec. 2), then formalizes the safety taxonomy (Sec. 3), analyzes safety challenges in FM-enabled robotics (Sec. 4), discusses alternative non-modular approaches (Sec. 5), and finally presents the modular safety guardrail architecture and its co-design opportunities (Sec. 6), where we explicitly characterize the architectural requirements for safe FM-robot deployment and introduce co-design principles that go beyond simple module stacking.

## 2. Roles of Foundation Models in Robotics

**FM as a Perception Module.** FMs are increasingly used as perceptual front ends in robotic systems, mapping raw sensory inputs such as images, depth, and language into high-level semantic representations (Ahn et al., 2022; Gorlo et al., 2025). For example, vision-language and multimodal FMs enable capabilities such as open-vocabulary object recognition (Liu et al., 2024a), scene understanding (Maggio & Carlone, 2025; Alama et al., 2025), and affordance prediction (Nasiriany et al., 2025), allowing robots to perceive previously unseen objects and environments without task-specific retraining (Gadre et al., 2023). By lifting perception from closed-set classification to semantic abstraction, FMs substantially expand a robot's operational scope, while also introducing new challenges related to uncertainty, grounding, and reliability in safety-critical settings (Ren et al., 2023; Huang et al., 2023; Kim et al., 2025).

**FM as a Reasoning Module.** Since the emergence of large language models (LLMs), a common integration of FMs in robotics is to use FMs as high-level semantic planners that interpret natural-language instructions and compose executable action sequences. Early systems grounded plan-

ning in predefined libraries of robot skills, translating user instructions into sequences of skill invocations. For example, given a skill set such as `move-to-<location>` or `grab-<object>`, an LLM can map an instruction like "bring me a bottle of water from the kitchen" into a structured plan. Representative studies include Code as Policies (Liang et al., 2022), ProgPrompt (Singh et al., 2023), PaLM-SayCan (Ahn et al., 2022), LLM-Planner (Song et al., 2023), and Alpamayo-R1 (Wang et al., 2025b).

**FM as an Action Module.** End-to-end vision-language-action (VLA) models extend this paradigm by adapting pretrained FM backbones to map images and language instructions directly to robot actions, unifying perception, grounding, and control in a single policy. RT-2 (Zitkovich et al., 2023) introduces an action-as-language approach by co-fine-tuning web-scale VLMs and representing actions as discrete autoregressive tokens. OpenVLA (Kim et al., 2024) similarly fine-tunes a Llama 2-based VLM into a 7B-parameter action generator trained on large-scale robot demonstrations. By contrast, the $\pi$-series (Black et al., 2025) preserves a pretrained VLM backbone (initialized from PaliGemma (Beyer et al., 2024)) and pairs it with a separate action expert for continuous control: $\pi_{0.5}$ (Black et al., 2025) extends $\pi_0$ (Black et al., 2024) via co-training on heterogeneous semantic supervision, while $\pi_{0.6}$ (Intelligence et al., 2025) scales the backbone (reported as 5B) and improves performance through on-robot experience. Gemini Robotics (Gemini Robotics Team et al., 2025) follows a similar end-to-end approach, fine-tuning a Gemini 2.0-based model to directly generate control commands.

## 3. Safety Definitions and Specifications for FM-enabled Robotics

This section introduces the *safety definitions* and associated *safety specifications* for FM-enabled robotic systems operating in human-centered, unstructured environments. We use *safety definitions* to describe conceptual categories of safety, and *safety specifications* to denote the explicit constraints that a robot must satisfy during operation.

Prior to the adoption of FMs, robotics safety primarily focused on *action safety*, which enforces low-level physical constraints. With FMs now integrated across the autonomy stack (Sec. 2), robots are increasingly deployed in unstructured, human-centered environments, inducing safety considerations that extend beyond physical execution. Thus, we organize safety into three complementary categories: *action safety*, *decision safety*, and *human-centered safety*.

❶ **Action Safety.** Action safety concerns maintaining a robot's physical execution within well-defined constraints, particularly in real-world environments (Hsu et al., 2023). Typical specifications include collision avoidance, adher-

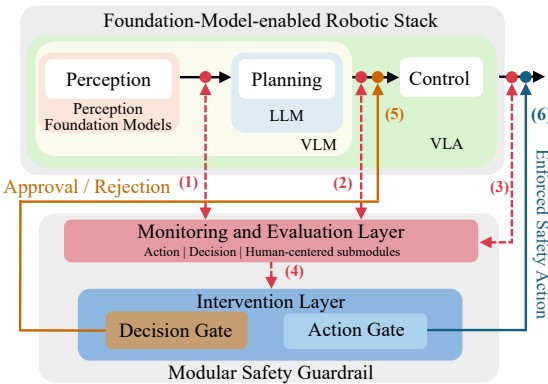

*Figure 2.* **Overview of one potential modular safety guardrail architecture.** It shows the architecture and information flow between the FM-enabled robotic stack (**top**) and the modular safety guardrail (**bottom**). Arrows **1–3** denote information flow from perception, planning, and control to the Monitoring and Evaluation Layer, which uses dedicated submodules for each safety dimension to generate risk signals for downstream modules. Risk indicators from planning and control are sent to the Intervention Layer (Arrow **4**), consisting of a *Decision Gate* that screens plans and triggers replanning upon rejection (Arrow **5**) and an *Action Gate* that enforces physical safety constraints on control commands. The Action Gate may apply last-resort safety filters to ensure only physically safe actions are executed (Arrow **6**).

ence to joint limits, and safe force or impedance modulation during interaction. These safety specifications are relatively well understood and are largely addressed through model-based control-theoretic approaches (Hsu et al., 2023; Wabersich et al., 2023; Hewing et al., 2020).

❷ **Decision Safety.** Decision safety generalizes action safety by incorporating semantic and contextual appropriateness in open-world settings. With LLMs integrated into robotic autonomy stacks, safety specifications must assess whether a robot's proposed behavior is appropriate with respect to open-domain knowledge and task semantics (Nakamura et al., 2025). For example, constraints such as *"a soft toy must not be placed on a hot stove"* (Gemini Robotics Team et al., 2025) and *"not pouring coffee too fast"* (Nakamura et al., 2025) illustrate decision-level safety requirements that cannot be captured by low-level constraints alone.

❸ **Human-centered Safety.** Human-centered safety concerns whether robot behavior is perceived by humans as predictable, understandable, and trustworthy over sustained interaction, such that physical, cognitive, and social risks remain acceptable (Hancock et al., 2011; Kress-Gazit et al., 2021). Even when action- and decision-level constraints are satisfied, misaligned expectations, non-stationary human adaptation, and miscalibrated trust can cause behavior to be perceived as unsafe (Sagheb et al., 2025; Peng et al., 2025).

**Intersections across dimensions.** These three dimensions are not mutually exclusive. Many real-world failures span

multiple dimensions simultaneously. For example, handing scissors to a child may pass action-level feasibility checks while violating decision-level appropriateness and human-centered norms; an action-only filter would approve such a handoff, while a semantic checker alone would lack real-time physical enforcement. Addressing such overlaps motivates the cross-layer and cross-module co-design described in Sec. 6.3.

## 4. Safety Challenges in FM-enabled Robotics

This section analyzes the primary safety challenges faced by FM-enabled robotic systems and their implications for the safety definitions introduced in Sec. 3, as illustrated in Fig. 1. We group these challenges by source, highlighting why modular runtime safety mechanisms are required beyond end-to-end learning alone.

### 4.1. Intrinsic FM and Robot Limitations

Many safety failures in FM-enabled robotics stem from intrinsic limitations of FMs as perceptual and reasoning components, as well as robots as physical systems. FM-enabled robotics inherits safety concerns already studied in perception, VLM, and LLM systems. However, embodiment introduces an additional safety burden: perception or reasoning errors can propagate through planning and control into unsafe physical behavior. In particular, epistemic unreliability in perception and reasoning primarily threatens *decision safety*, while hard physical constraints and modeling mismatch threaten *action safety*.

**Epistemic unreliability under distribution shift.** Despite rapid progress, VLM/VLA systems remain brittle under distribution shift, exhibiting hallucination, weak spatial grounding, and poorly calibrated uncertainty (Liu et al., 2024b; Chakraborty et al., 2025b; Chen et al., 2024a). For generative models, reliability failures extend beyond input out-of-distribution (OOD) detection to the trustworthiness of generated outputs, which is not well captured by conventional uncertainty measures (Ovadia et al., 2019; Xu & Ding, 2025). These failures can propagate to downstream decisions and induce unsafe behaviors.

**Hard physical constraints and long-tail failures.** Action safety is constrained by hardware limits, unmodeled dynamics, and environment-imposed constraints (Hsu et al., 2023). Safety-critical physical failures are inherently rare and dominated by worst-case interactions (Brunke et al., 2022; Kojima et al., 2025; He et al., 2025), creating a mismatch between statistical learning objectives and hard constraint satisfaction. Explicit low-level enforcement therefore remains necessary for open-world deployment.

### 4.2. Adversarial and Worst-case External Factors

A second source of risks comes from adversarial manipulation and worst-case external factors, which threaten *decision safety* via perception- and language-level attacks and *action safety* via disturbances that violate nominal assumptions. Multimodal FM-enabled control pipelines expose expanded attack surfaces (e.g., prompt- and perception-level manipulations) that can bypass intended safeguards (Jones et al., 2025; Robey et al., 2025; Xing et al., 2025), while real-world operation entails disturbances and unexpected contacts that are difficult to model exhaustively. This motivates runtime safety mechanisms that are decoupled from FM-based decision modules, such as certified control or safety-filtering layers that enforce physical constraints independently of high-level reasoning (Hsu et al., 2023; Ames et al., 2019).

### 4.3. Open-world Deployment

FM-enabled robots increasingly operate in open-world settings where tasks, constraints, and contexts are compositional and effectively unbounded (Firoozi et al., 2025). This shift makes exhaustive specification, testing, and offline validation infeasible. Such deployment induces task-space explosion and long-tail safety risks, where rare but critical failure modes dominate overall risk (He et al., 2025; Angelopoulos et al., 2023). Ensuring safety therefore requires lifecycle-level strategies that span prevention, runtime monitoring and mitigation, and post-incident recovery, rather than relying solely on pre-deployment training or static rules (Kojima et al., 2025; Tan et al., 2025).

### 4.4. Human Complexity and Mutual Adaptation

Human-centered safety challenges arise from the interactive, subjective, and evolving nature of human behavior and expectations, primarily defining *human-centered safety* while also feeding back into *decision safety* through ambiguous intent and dynamic preference shifts. Language-enabled interaction introduces ambiguity in user intent (Ren et al., 2023; Santos et al., 2025; Mehta et al., 2024), and human-perceived safety is inherently subjective, context-dependent, and personalized, making it difficult to encode as fixed constraints or universal objectives (Santos et al., 2025; Mehta et al., 2024). Even when action- and decision-level constraints are satisfied, mutual adaptation, shifting trust, and evolving social norms can cause robot behavior to be perceived as unsafe (Tan et al., 2025), rendering static thresholds and uncertainty-only formulations insufficient for sustained human–robot interaction.

## 5. Alternative Views

This section organizes existing safety literature around alternative views on *where* safety should be placed within

FM-enabled robotic systems. These views include embedding safety considerations within end-to-end models, as well as applying isolated external add-ons adapted from perception, LLM, and VLM safety studies, such as alignment, uncertainty estimation, semantic filtering, and preference modeling. Each view addresses a subset of the safety challenges in Sec. 4, but none provides comprehensive protection across action, decision, and human-centered risks in open-world deployment. This analysis motivates the need for the modular guardrail architecture introduced in Sec. 6.

### 5.1. View A: Safety Should be Embedded in the Model

One alternative view argues that safety should be internalized directly within the FM or base policy, such that safe behavior emerges by construction rather than external constraints. This view is supported by post-training alignment methods, such as instruction-tuning and reinforcement learning from human feedback (Ouyang et al., 2022), which can substantially shift model behavior toward user intent and reduce undesirable outputs. Related efforts, such as principle-based alignment (e.g., Constitutional AI (Bai et al., 2022; Gemini Robotics Team et al., 2025)) and recent safety pre-training proposals (Maini et al., 2025), further advocate treating safety as a first-class training objective so that internal representations and decision rules are safety-aware. **What it cannot cover?** Embedding safety within the model does not establish a non-bypassable runtime boundary between high-level decision-making and physical execution. Even well-aligned models remain vulnerable to distribution shift, novel hazards, and unforeseen open-world interactions. While increasingly capable end-to-end systems may reduce reliance on modular safeguards in controlled settings, we argue that externally enforceable modular guardrails remain necessary for open-world deployment with evolving constraints and long-tail failures. When failures occur, there is no external mechanism to prevent unsafe actions from being executed. As a result, model-internal safety alone cannot guarantee action-level safety at deployment time.

Recent VLA systems also suggest that modular adaptation can remain useful within otherwise end-to-end pipelines. For example, RL Token (Xu et al., 2026) adapts behavior through lightweight residual modules rather than fully retraining the entire VLA. Similarly, adapting modular safety guardrails to deployment-specific requirements may often be more practical than fully fine-tuning large foundation models for every downstream safety domain.

### 5.2. View B: Safety Should be Provided by External Add-ons (Single-perspective or Narrow Subset)

A second view places safety outside the FM, implemented through external modules that monitor, filter, or constrain system behavior at runtime. Most existing approaches adopt a single perspective, addressing only one safety dimension or a narrow subset rather than providing integrated coverage across all three safety dimensions.

**View B1: Action-level add-ons.** Action-level approaches include safety standards (ANSI/RIA, 2012; ISO, 2011; Jacobs & Virk, 2014) and control-theoretic safety filters (Margellos & Lygeros, 2011; Fisac et al., 2019; Ames et al., 2019; Bastani, 2021; Wabersich & Zeilinger, 2018) that intervene during execution to enforce physical constraints. These methods are effective for ensuring collision avoidance, constraint satisfaction, and physical feasibility. **What it cannot cover?** Action-level mechanisms cannot reason about semantic hazards, task-level mistakes, or harmful intent, nor can they capture human-contextual risks. They also rely on predefined unsafe sets and sufficiently accurate models of system dynamics, assumptions that often break down for FM-enabled robots operating in unstructured environments (Hsu et al., 2023; Bajcsy & Fisac, 2024).

**View B2: Decision-level add-ons.** Decision-level approaches attempt to detect unsafe plans or commands before execution, using learned world models (Nakamura et al., 2025; Seo et al., 2025; Agrawal et al., 2025), LLM-based judges (Gu et al., 2024; Duan et al., 2024; Yang et al., 2025; Gao et al., 2025; Khan et al., 2025; Ravichandran et al., 2025; Jindal et al., 2025) or uncertainty estimation techniques (Ren et al., 2023; Liang et al., 2024; Sun et al., 2024b;a; Wang et al., 2025a; Karli et al., 2025). These methods directly target decision-level safety by filtering or modifying high-level outputs. **What it cannot cover?** They cannot guarantee physical safety during execution (Bajcsy & Fisac, 2024), are vulnerable to hallucinations and adversarial manipulation when LLMs are involved (Chen et al., 2024b; Gao et al., 2024; Xing et al., 2024; Xu et al., 2025; Jones et al., 2025; Lechner et al., 2023; Everett et al., 2021), and often assume access to complete or correct safety specifications. Additionally, their performance degrades significantly under distribution shift, limiting reliability in real-world deployment (He et al., 2025).

**View B3: Human-centered add-ons.** Human-centered approaches rely on intent inference, preference learning, trust modeling, and feedback-based adaptation to align robot behavior with human expectations (Peng et al., 2025; Dixit et al., 2023; Chakraborty et al., 2025a; Salzmann et al., 2020; Chen et al., 2025; Pandya et al., 2025). These methods improve interaction quality and personalization. **What it cannot cover?** They cannot provide hard safety guarantees or enforcement to prevent unsafe actions when misalignment occurs, as human intent is inherently ambiguous, preferences change over time, multi-human environments introduce conflicting constraints (Shi et al., 2025).

**Key Takeaway**

Neither model-internal safety nor single-perspective external add-ons are sufficient in isolation. Internal alignment improves typical behavior, but cannot replace non-bypassable enforcement at execution time, while external add-ons usually address only one safety dimension. Ensuring safety in FM-enabled robotic systems, therefore, requires a modular safety guardrail architecture that provides integrated coverage across action-level, decision-level, and human-centered risks, as developed in Sec. 6. These insufficiencies motivate the architectural requirements (Sec. 6.1) and co-design principles (Sec. 6.3); stacking such individually insufficient mechanisms would inherit the same gaps.

## 6. Modular Safety Guardrails

### 6.1. Definition and Design Principles

We argue that architectural modularity is necessary for the safe and reliable deployment of FM-enabled robotic systems. As discussed in Sec. 4, such systems face non-stationary and hard-to-predefine safety specifications, compounded uncertainty from both the environment and FMs, and rare but catastrophic safety failures. Addressing these challenges requires safety mechanisms that are (i) **independently verifiable and updateable** as requirements evolve, (ii) **composable** across complementary failure modes spanning physical, semantic, and human-interaction contexts, and (iii) **non-bypassable at execution time**.

We capture these requirements with a *modular safety guardrail*: a safety layer decoupled from upstream autonomy components that supports independent updates, composition of heterogeneous mechanisms, and enforceable closed-loop intervention. Here, we use *guardrails* as an umbrella term for modular safety mechanisms consisting of (i) *monitoring* components that evaluate safety conditions and (ii) *intervention* components that constrain or override actions when violations are detected, as introduced in Sec. 6.2.

**Definition 1 (Modular safety guardrail).** A modular safety guardrail is a non-bypassable, execution-time safety architecture that mediates all execution-relevant proposals (e.g., perceptions, plans, and commands) from upstream autonomy components before they reach the robot's physical execution layer. Operating in the closed-loop robot-environment-human system, it monitors risk and intervenes at runtime to prevent unsafe behavior across physical, semantic, and interaction-level safety dimensions.

A modular safety guardrail is characterized by two forms of modularity: *external* and *internal*.

**External modularity.** An external guardrail is designed to be operationally independent of the upstream FMs it monitors. It should not share parameters, training objectives, or optimization procedures with those FMs, and should minimize statistical coupling (e.g., shared pretraining corpora or safety-annotation data) that could induce correlated errors. This independence enables the guardrail to function as an auditable, verifiable safety authority rather than inheriting the same uncertainty and failure modes as the guarded FM.

**Internal modularity.** Internal modularity decomposes the guardrail into specialized submodules with explicit interfaces and non-overlapping safety responsibilities, each targeting distinct failure modes and time-scale requirements (e.g., perception trust assessment, decision-level semantic or intent screening, and action-level constraint enforcement). Submodules may consume different autonomy-stack signals, yet remain independently testable, replaceable, and updateable without retraining the entire guardrail. This structure limits cross-dimension error propagation and supports robust enforcement under evolving specifications and real-time constraints.

### 6.2. Modular Safety Guardrail Architecture

The modular safety guardrail decomposes safety enforcement into two functionally distinct layers (Fig. 2, bottom): a *Monitoring and Evaluation Layer* that **assesses risk** across autonomy-stack outputs, and an *Intervention Layer* that **enforces safety** through two complementary submodules: a *decision gate* and an *action gate*. This two-layer architecture instantiates the internal modularity principle of Sec. 6.1, while external modularity is preserved through the guardrail's operational independence from the upstream FM stack. The further subdivision of the Intervention Layer into decision and action gates represents a fine-grained decomposition of the same principle. Together, these layers address the complementary failure modes across the three safety dimensions identified in Sec. 3.

#### 6.2.1. MONITORING & EVALUATION LAYER

The Monitoring and Evaluation Layer performs independent risk assessment across the autonomy stack, aiming to detect potential safety violations before they propagate downstream. It observes execution-relevant outputs from perception, planning, and control through channels decoupled from the primary autonomy pipeline. These channels may include auxiliary sensors for cross-validating perception (Antonante et al., 2023a;b), FM-based evaluators (e.g., critic or red-teaming agents) for assessing high-level plans (Elhafsi et al., 2023; Sinha et al., 2024), and risk-aware control-theoretic monitors for detecting physical constraint violations (Frank et al., 2024; Lyu et al., 2023). To help reduce shared failure modes between the system being evaluated and the evaluator, this layer interfaces with explicit, execution-relevant

representations (e.g., regions of interest, candidate plans and trajectories, and control commands) and does not rely on internal latent embeddings unless they are explicitly exposed.

The layer can produce risk signals spanning all three safety dimensions (through dedicated submodules). Not all safety properties are equally monitorable or formally specifiable. Low-level physical constraints, such as joint limits, collision margins, or velocity bounds, are often amenable to runtime monitoring and enforcement. In contrast, higher-level semantic and human-centered safety properties may only admit approximate, probabilistic, or human-in-the-loop evaluation, and exact online verification may be computationally intractable in real-time embodied settings. The proposed modular architecture does not resolve these formal limitations, but instead decomposes safety into subproblems with different monitoring and computational requirements. For action safety, it flags violations of joint limits, collision margins, or dynamic feasibility. For decision safety, it checks semantic consistency and contextual appropriateness, including FM-specific issues such as hallucination or adversarial vulnerability. For human-centered safety, it monitors predictability, preference alignment, and trust-related indicators. These signals are passed independently to the Intervention Layer, which executes non-bypassable mitigation and coordinates enforcement across dimensions via co-design (Sec. 6.3).

### 6.2.2. INTERVENTION LAYER

Because robots are physically embodied, recognizing risk is not enough: unsafe proposals must be blocked or modified before reaching the actuators. The *Intervention Layer* provides this non-bypassable, execution-time authority by applying concrete mitigations when monitored risk exceeds acceptable thresholds, through two complementary mechanisms: a planning-level *decision gate* and an execution-time *action gate*.

This two-gate design reflects that safety risks arise at different semantic levels and time scales. High-level failures (e.g., misinterpreted intent, unsafe semantics, and norm violations) should be intercepted before execution commits the robot to an inappropriate course of action. Conversely, even semantically appropriate plans may become unsafe under disturbances, tracking error, state-estimation drift, or unmodeled contacts, requiring an action gate as the last line of defense. Separating these roles localizes responsibility and supports principled conservatism allocation: reject plans only when necessary, and otherwise enforce safety through minimal execution-time modification.

**Decision Gate.** The decision gate operates at the planning level, screening candidate plans using aggregated risk signals from the Monitoring and Evaluation Layer (Fig. 2, solid brown arrow (5)). It targets violations of decision safety and

human-centered safety in FM-generated plans. We design the gate as a filter that is explicitly decoupled from plan generation, preserving a clear safety authority boundary between proposing actions and approving them for execution. When risk exceeds acceptable thresholds, the gate blocks the plan and triggers replanning or user clarification. By filtering semantically or socially unsafe plans upstream, the decision gate prevents the action gate from compensating for fundamentally inappropriate intent, reducing unnecessary stops and overly conservative execution.

**Action Gate.** The action gate operates at the execution level, enforcing physical safety by constraining or modifying low-level control commands before they are applied to the robot (Fig. 2, solid navy arrow (6)). It enforces action safety via shielding, trajectory adjustment, or projection onto safe sets, with constraints parameterized by monitoring outputs such as uncertainty or trust. Unlike the decision gate, which reasons over plan content, the action gate provides non-bypassable physical enforcement regardless of how plans are generated. Architecturally, it is the last line of defense: even after a plan is approved, it ensures executed commands remain within acceptable physical bounds under disturbances and residual upstream failures.

**Safety Assurance.** The safety assurances of the proposed architecture come from enforcing a clear safety authority boundary rather than from assuming any single model is correct. Concretely, (i) non-bypassability ensures all execution-relevant proposals pass through the intervention layer before reaching the actuators; (ii) external modularity/operational independence enables independent auditing and verification; and (iii) internal modularity allows heterogeneous safety mechanisms to be verified, updated, and composed without retraining the full stack. Together, these properties provide enforceable runtime safety envelopes (via the action gate) and upstream rejection of semantically or socially unsafe plans (via the decision gate), yielding systematic coverage across action, decision, and human-centered safety.

### 6.3. Co-Design Opportunities Enabled by Modular Guardrails and Deployment Examples

The modular safety guardrail architecture provides an enforceable foundation for comprehensive safety across action, decision, and human-centered dimensions. Beyond this foundation, the architecture also exposes a new space for co-design that can make safety enforcement faster, less conservative, and more graceful in practice. We view possible co-design systematically along two axes: (i) between layers (Monitoring and Evaluation and Intervention Layers) and (ii) within a layer (coordination among modules inside the Intervention Layer). We highlight two key opportunities.

**Representation alignment: Co-design between layers.** Representation alignment concerns the interaction between

layers, i.e., how safety-relevant information produced by the Monitoring and Evaluation Layer is represented so that the Intervention Layer can make more informed interventions. The core principle is representation compatibility: monitoring outputs should preserve uncertainty and risk structure that downstream enforcement primitives can act on directly, rather than collapsing them into lossy scalar scores. For example, expressing perception uncertainty as a spatially grounded set (e.g., ellipsoidal pose uncertainty, occupancy tubes, and workspace-indexed risk fields) allows the action gate to tighten constraints directionally or locally where risk is concentrated, instead of applying uniform conservatism everywhere. In this way, co-design at the layer interface turns "risk assessment" into actionable, enforcement-ready inputs that support precise intervention.

**Conservatism allocation: Co-design between modules.** Conservatism allocation concerns interaction among modules inside the Intervention Layer, especially between the decision gate and the action gate. If each module applies its strictest criterion independently, conservatism stacks, often producing infeasible behavior (premature plan rejection, excessive projection, or unnecessary stoppages). Co-design enables coordinated strictness: the decision gate can approve plans conditionally on whether the action gate can enforce the induced margins in real time, and can escalate to rejection or human clarification only when the action gate reports infeasibility. This shifts conservatism to the module that can enforce it most precisely, preserving task progress while maintaining safety.

Together, these two strategies clarify how the proposed architecture enables more than modular enforcement: co-design across layers improves what information is enforced, while co-design within layers improves how enforcement authority is exercised without compounding conservatism.

Next, we provide three[1] deployment examples to illustrate how the proposed modular safety guardrail interfaces with different FM configurations. Each example serves as an existence proof of a concrete failure mode that cannot be addressed by model-internal or single-perspective safety alone, and demonstrates how cross-layer or cross-module co-design enables effective resolution in practice.

**Example A: Language-Instructed Household Robot (LLM/VLM-based Planner).** A household robot follows natural-language instructions (e.g., "clean up the living room") using an LLM planner to generate skill sequences (Singh et al., 2023; Ahn et al., 2022). Such planners can produce unsafe or ill-posed plans due to semantic misinterpretation, ambiguous intent, or hallucinated affordances (e.g., discarding items to be preserved, placing a hot pan on

---

[1]Due to space considerations, one representative example is included in the main text, with two additional ones in Appendix A.

wood, or assuming a drawer is open). The monitoring layer checks semantic consistency via LLM-as-a-Judge (Elhafsi et al., 2023; Sinha et al., 2024), quantifies uncertainty with conformal prediction (Ren et al., 2023), verifies constitutional rules (Sermanet et al., 2025; Jindal et al., 2025), and evaluates human-centered risks (predictability and intent alignment). The decision gate rejects or requests clarification when risk or uncertainty is high, while the action gate enforces physical constraints through trajectory projection onto safe sets (Fisac et al., 2019).

*A co-design example of representation alignment and conservatism allocation:* While carrying a hot pan from the stove to the counter, the robot observes a child entering and moving toward its planned path. Without co-design, the decision gate may apply a fixed semantic rule (e.g., "hot object near child") and reject the plan, freezing the robot mid-motion while holding a hazard. With co-design, the monitoring layer converts this semantic hazard into the same object the action gate can enforce: a time-varying keep-out region (or occupancy tube) for the child, with a radius scaled by thermal risk rather than collision-only risk (representation alignment). The decision gate then allocates conservatism by approving continuation whenever the action gate can certify feasibility of maintaining separation under these means (conservatism allocation), and escalating only when it cannot. In execution, the action gate enlarges clearance (e.g., 1.0 m for thermal hazards vs. 0.3 m for collision-only), reduces speed, and reroutes to a farther placement location, allowing the robot to safely complete the placement instead of defaulting to a brittle stop.

## 6.4. Scope and Limitation

The modular safety guardrail is not a universal solution to safety in FM-enabled robotics. It is not intended to make the FMs intrinsically safer or to guarantee globally optimal decisions. Rather, it provides an enforceable runtime layer that detects unreliable outputs and mitigates their consequences, complementing offline retraining or fine-tuning, which cannot be relied on during online execution. We do not claim that modular architectures will always outperform alternative safety approaches (e.g., end-to-end safety), but rather that externally enforceable runtime safeguards remain necessary for practical open-world deployment. In this sense, the guardrail improves deployability by expanding safety enforcement beyond physical constraints, enabling intervention on contextually inappropriate high-level decisions while retaining action-level gating as the last line of defense against physical harm.

Another motivation for modularity comes from heterogeneous complexity and verification requirements across safety dimensions. Some safety properties may admit formal verification or tractable runtime checking, while oth-

ers involving open-world semantics, human intent, or long-horizon reasoning may be computationally intractable or difficult to fully specify in a decidable form. The proposed modular architecture allows different safety mechanisms to operate under different computational and formal assumptions, enabling practical real-time enforcement without requiring a single globally verifiable end-to-end safety model.

Another practical limitation is that the effectiveness of the guardrail depends on the independence of its safety signal from the FM-based autonomy stack it monitors. When the guardrail itself relies on FMs, such "independence" may be hard to ensure in practice because models from different sources can share training data, design choices, or evaluation pipelines (e.g., LLMs/VLMs like Qwen (Bai et al., 2025), Llama (Touvron et al., 2023), Gemini (Comanici et al., 2025), GPT (Achiam et al., 2023) or VLAs like Open-VLA, GR00T (Bjorck et al., 2025), Gemini Robotics). This overlap can induce correlated failure modes, weakening FM-based guardrails as independent checks. These challenges motivate clearer notions of relative or operational independence, including metrics that quantify statistical dependence or shared failure risk, to characterize when FM-based guardrails provide meaningful safety benefits.

A further practical consideration is runtime execution. FM-based guardrails introduce computational overhead, and we view this as a necessary cost of test-time scaling for safety (Kwok et al., 2025; Wu et al., 2025), where additional computation is allocated to safety-critical monitoring and intervention. Although modular architectures can mitigate latency through parallel safety intervention between the decision and action gates, the decision gate may still incur delays from large-scale FMs needed to reason about complex causal and contextual relationships. Practical deployment therefore requires cross-module co-design that accounts for gate asynchrony, including fallback mechanisms that allow the action gate to apply conservative adjustments while the decision gate completes its reasoning.

## 7. Call to Action & Conclusion

We argue that safety for FM-enabled robotics must be approached as a system property, requiring explicit mechanisms that jointly address action, decision, and human-centered risks. We advocate modular safety guardrails as a practical architectural foundation to decouple safety authority from any single model, support independent auditing and updates, and enable robust deployment across tasks and platforms. Our goal is not to propose a complete safety solution, but to identify architectural conditions that are necessary for any safety mechanism to scale to open-world FM-enabled robotic systems. While grounded in robotics, we view the safety challenges and architecture described in this paper as broadly applicable to embodied, agentic AI

systems that couple FM reasoning with real-world action under uncertainty. We invite the community to help turn this architectural position into a practical, shared safety stack along two complementary directions.

First, we call for a broader ecosystem of composable guardrail modules. Beyond basic monitoring and intervention components, there is a large design space for modules that target specific failure modes, uncertainty sources, and human interaction contexts. To enable reuse across platforms and tasks, such modules should expose clear interfaces and semantics. Progress here also depends on evaluation: we encourage the development of comprehensive benchmarks that explicitly test action, decision, and human-centered safety, including rare but catastrophic scenarios that are systematically underrepresented in today's datasets.

Second, we call for principled co-design within the guardrail architecture. Co-design goes beyond assembling modules; it requires specifying what safety-relevant information is produced (e.g., risk, uncertainty, and constraint structure), how it is represented, and how it flows bidirectionally across the autonomy stack. Such co-design allows upstream components to learn to avoid repeatedly proposing actions that downstream gates will reject or heavily modify. With appropriate co-design, safety can be faster to enforce, less conservative in practice, and more robust to deploy, by allocating conservatism where it is needed based on a systematic view instead of accumulating it everywhere.

We hope this position paper serves as a starting point for a shared research agenda: to standardize interfaces, expand modular guardrail capabilities, and develop co-design principles that yield composable, updateable, and deployment-ready comprehensive safety mechanisms for FM-enabled robotic systems and related embodied AI platforms.

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

# A. Additional Deployment Examples for Modular Guardrails and Co-Design

This appendix provides two additional scenarios expanding on Sec. 6.3, illustrating how modular safety guardrails integrate with different FM-enabled robotics stacks and can be extended through cross-layer and cross-module co-design.

**Example B: Open-Vocabulary Mobile Manipulation (Perception FM + Classical Planning/Control).** A warehouse robot uses CLIP-based open-vocabulary detection (Gadre et al., 2023), along with a segmentation model e.g., SAM (Kirillov et al., 2023), for object identification, combined with classical motion planning and control. During a pick-and-place task, the perception FM may misidentify objects, for instance, confusing a fragile glass container with a plastic one, leading to inappropriate grasping force, or hallucinating object presence in cluttered scenes due to visual ambiguity. The Monitoring and Evaluation Layer generates trustworthiness scores by cross-validating detections against depth sensors and tracking consistency across frames; discrepancies (e.g., depth discontinuities inconsistent with detected object geometry) indicate potential misidentification. OOD detection methods (Farid et al., 2022) flag high-uncertainty cases such as novel object categories absent from training data or ambiguous boundaries caused by occlusion. Since no plans are produced by the FM-based component in this setting, the decision gate is optional and primarily used to prevent plans from relying on low-confidence perception outputs. The action gate provides execution-time enforcement (e.g., collision avoidance constraints), ensuring physical safety even when upstream perception is unreliable.

*A co-design example of representation alignment:* The key is to give the monitoring layer a representation that enables more informed downstream enforcement: instead of a scalar confidence, it outputs a pose-uncertainty ellipsoid that preserves the magnitude and direction of localization error. In low light, a low scalar score would otherwise force a blunt stop/go decision. With the ellipsoid, the action gate can enforce safety more precisely by tightening margins anisotropically (e.g., 8 cm along the major axis, 2 cm along the minor axis), and in future steps, the decision gate approves the same plan only if these ellipsoid-induced constraints remain feasible in the current workspace. This richer representation lets the guardrail be cautious where needed without resorting to uniform conservatism.

**Example C: Dexterous Manipulation (End-to-end VLA Policy).** A manipulation robot uses an RT-2-style VLA policy (Zitkovich et al., 2023) that maps visual observations and language instructions directly to control commands. This end-to-end design introduces distinct risks: perception or grounding errors can immediately translate into unsafe motion; the policy may hallucinate object locations under visual ambiguity or out-of-distribution scenes; and per-step "reasonable" commands can still accumulate into an unsafe path in clutter, gradually eroding clearance or steering the arm toward joint/workspace limits. Because the policy exposes few intermediate representations, failures are difficult to attribute (e.g., perception vs action prediction). The monitoring layer estimates action-level risk via token-level uncertainty from prediction entropy (Karli et al., 2025) and cross-validates scene understanding with an independent perception check against workspace constraints to flag hallucinations. Since no explicit plan is produced, the decision gate is bypassed and the action gate becomes the primary safeguard, projecting commands onto safe sets (Fisac et al., 2019; Ames et al., 2019) near collision or limit boundaries and triggering a fallback (e.g., controlled retraction) in high uncertainty. This setup mainly enforces action safety, with limited decision safety through uncertainty signals.

*A co-design example of representation alignment and conservatism allocation:* In the same VLA setting, the robot is told to "pick up the red cup" on a cluttered table near a glass vase; token-level entropy is high because the policy is unsure which candidate object to target. Without co-design, entropy is thresholded as a scalar, forcing a binary choice: execute with nominal constraints or fall back. With co-design, the monitor localizes uncertainty using attention/saliency and maps it into workspace coordinates as a spatial risk field, which the action gate uses to tighten margins and slow down only near high-risk regions. This turns uncertainty into localized, enforceable caution, enabling safe grasping without a hard stop.

