# OpenReview forum: "Position: Modular Safety Guardrails Are Necessary for Foundation-Model-Enabled Robots in the Real World"
_ICML.cc/2026/Position_Paper_Track — ICML 2026 Position Paper Track regular_

### Official Review · Reviewer_TXRr · 2026-03-11

**Significance:** 3
**Argument Clarity:** 2
**Rating:** 5
**Confidence:** 4

**Questions:**

Could you please try and address the Weaknesses pointed out above?

**Alternative Views Section:**

Yes

**Compliance With Llm Reviewing Policy A Conservative:**

Affirmed.

**Discussion Potential:**

3

**Final Justification:**

The rebuttal addressed your main concerns,

**Paper Summary:**

This position paper considers the question of integrating foundation models (FMs) into robotics and offers a characterization of FM-enabled robot safety along three dimensions: action safety (physical feasibility and constraint compliance), decision safety (semantic and contextual appropriateness), and human-centered safety (conformance to human intent, norms, and expectations).
The authors argue that architectural modularity is necessary for the safe and reliable deployment of FM-enabled robotic systems.

**Position:**

Yes

**Position In Title:**

Yes

**Related Work:**

3

**Strengths And Weaknesses:**

Strengths

1) The paper is well-informed and well-motivated and the related literature is reviewed extensively and in detail.

2) While I do not share entirely the analysis provided in the paper, I think the authors make their case rather convincingly and this paper is thought-provoking and might be a nice addition to the conference programme.


Weaknesses

1) The authors say that "classical robotics safety often assumes fixed and predefinable constraints (e.g., geometric collision bounds),
whereas FM-enabled robots operate in open-ended environments where hazards are context-dependent and specifications evolve".
But this is true not only for FM-enabled robots, but for the deployment of robots in real-life, safety-critical environments in general. It is true that FM-enabled robots might be more prone to this kind of issues, but it is definitely not just a feature of FM-enabled robots.

2) The three notions of safety identified by the authors (action, decision and human-centered) are presented almost as alternatives. But I would argue that in general they are not, and you might have intersections between these 3 notions. It is not clear what is the authors' stand on this, it would be helpful to clarify it.

3) When discussing alternative views, the authors say that "Even well-aligned models remain vulnerable to distribution shift, novel hazards, and unforeseen open-world interactions."
But in a sense this also applied to the monitoring approach they champion, as monitors might also need to be updated when deployed in new environments.

4) The authors identify 3 different notions of safety but then they use a unique monitor to check for safety. If they suggest that the 3 notions of safety are distinct, I'd expect a unique monitor for each of them.

**Support:**

3

---

> ### Author Rebuttal · Authors · 2026-03-31
>
> **Response to Reviewer TXRr:**
>
> We appreciate the reviewer’s feedback and address each concern below:
>
> **Overstated Distinction of FM-Enabled Robot Safety Problem (W1)**
>
> We agree that context-dependent and evolving safety requirements are not unique to FM-enabled robots, but are a broader challenge in real-world, safety-critical robotics. We will clarify this point in Sec. 4. Our emphasis is that FM-enabled robots often amplify these challenges in two ways. First, FM integration is pushing robots into less structured, more open-world settings with richer human interaction, where long-standing issues such as distribution shift, uncertainty, and complex human behavior become more pronounced. Second, FMs introduce additional risks tied to their own failure modes under embodiment. For example, beyond classical perception errors, FM-enabled robots may hallucinate, over-interpret instructions, and then justify or act on those mistaken beliefs (e.g., mistaking medication for a beverage and offering it as a drink to a thirsty person, or interpreting "clean up" as discarding important items). These FM-specific vulnerabilities do not replace broader robotics safety concerns, but they do make the safety problem more severe and motivate dedicated treatment.
>
> **Clarifications on Alternative and Our Positions: Intersections Across Safety Dimensions (W2)**
>
>  We would like to clarify that the three safety dimensions are not intended as separate alternatives, but can have intersections between two or all of them. We emphasize that real-world safety failures can span multiple dimensions simultaneously, motivating our co-design approach that jointly assesses and enforces risk signals across all three dimensions (Sec. 6.3), as illustrated by the following example. . Consider a robot instructed to "hand over the scissors" to a child. This may pass action-level safety checks, yet is contextually inappropriate and socially unacceptable. A purely action-level filter would approve this, while a purely semantic checker might flag it but lack real-time enforcement to physically modify the handoff motion. Our modular guardrail addresses such overlaps through cross-layer/cross-module co-design (Sec. 6.3). We will further clarify this in the final version of Section 3.
>
> **Guardrail Reliability Under Correlated Failures (W3 & AW59-Q3, Q5)**
>
> We agree that monitoring modules can also fail under distribution shift or in open-world scenarios (Sec. 6.4). Our point is not that monitors are exempt from open-world failure, but that modular guardrails make such failures less likely to propagate unchecked.
> External modularity reduces correlated failure by separating components across parameters, objectives, and training procedures (Sec. 6.1). Internal modularity further reduces shared failure modes by combining different mechanisms: for instance, an FM-based semantic monitor may fail, while a non-FM action gate such as a CBF-based controller can still prevent unsafe execution. Additional modules such as uncertainty estimation or OOD detection can also flag unreliable inputs before they reach control.
> Thus, modularity does not solve the problem completely, but it adds layered opportunities for detection, containment, and safe fallback, including safe stopping when consistent safe actions cannot be identified (Examples in Appx. A).
>
>
> **Distinct Monitoring Mechanisms per Safety Dimension (W4 & AW59-Q4)**
>
> We would like to clarify that distinct safety dimensions indeed call for distinct monitoring mechanisms, and that this is precisely what our internal modularity provides. As described in Section 6.2.1, the Monitoring and Evaluation Layer assesses all three dimensions through dedicated submodules: auxiliary sensors and control-theoretic monitors for action safety, FM-based evaluators for decision safety, and intent and preference monitors for human-centered safety. These heterogeneous risk signals are observed through separate, decoupled channels and trigger different intervention behaviors depending on the safety dimension they target. When dimensions overlap, each submodule passes its risk signal independently to the Intervention Layer, which coordinates enforcement through co-design. We refer the reviewers to Example A in Section 6.3 and Appx. A. We guess it is Figure 2 which causes confusion. We will incorporate these clarifications and update Figure 2 (including the caption) to clarify this.

---

> > ### Author Rebuttal · Reviewer_TXRr · 2026-04-02
> >
> > The authors addressed most of my concerns. I'll revise my score accordingly.

---

### Official Review · Reviewer_AW59 · 2026-03-12

**Significance:** 3
**Argument Clarity:** 2
**Rating:** 4
**Confidence:** 4

**Questions:**

1. In section 6, authors first introduce the internal and external modularity, and then propose two layers as safety guardrails. What is the relationship between internal/external modularity and two-layer guardrail? Is the first layer used as the external module?
2. How can the proposed architecture address the real-time constraints which need fast safety response? The latency and monitoring cost of the proposed architecture are not discussed in this paper.
3. If guardrails use LLMs or VLMs, how can correlated failures be avoided? The paper stresses external modularity and independence between guardrails and foundation models. But this question is not answered clearly.
4. The monitoring layer can produce risk signals spanning all three safety dimensions, which is a complex operation. How signals from perception, planning, and control should be combined in this process?
5. What if the proposed safety guardrails fail? The paper focuses on failures of the autonomy stack but does not analyze failures of the guardrail system. What happens if the monitoring layer is wrong? Can the guardrail create unsafe deadlocks or oscillations? These are important questions authors do not answer.

**Alternative Views Section:**

Yes

**Compliance With Llm Reviewing Policy A Conservative:**

Affirmed.

**Discussion Potential:**

3

**Ethics Review Area:**

["Other Expertise"]

**Final Justification:**

The authors addressed most of my main concerns. As such I increased rating by one.

**Paper Summary:**

This position paper argues that the deployment of FM-enabled robots introduces new safety challenges that extend beyond traditional physical safety in robotics. The authors identify three key safety dimensions, i.e. action safety, decision safety, and human-centered safety. It also argues that existing approaches such as model-internal safety mechanisms or single-layer external safeguards cannot adequately address these risks in open-world environments. To address this limitation, the paper proposes a modular safety guardrail architecture consisting of a Monitoring and Evaluation layer that assesses risks across perception, planning, and control, and an Intervention layer that enforces safety through a decision gate for filtering unsafe plans and an action gate for enforcing physical constraints during execution. The authors argue that this modular design enables independent verification, flexible updates, and coordinated safety enforcement across multiple levels of the robotic autonomy stack, providing a scalable foundation for safe deployment of FM-enabled robotic systems in real-world environments.

**Position:**

Yes

**Position In Title:**

Yes

**Related Work:**

3

**Strengths And Weaknesses:**

Strength:
1. The paper emphasizes that safety must be treated as a system property, not as a feature of a single model. This perspective aligns with broader trends in AI safety engineering and system assurance.
2. The paper introduces several important system design principles, including external and internal modularity, representation alignment, conservatism allocation. These concepts provide useful design guidelines for building safe robotic systems.
3. The paper addresses a rapidly emerging issue, which includes the integration of FM in robotics, the development of VLA, and deployment of general-purpose robots. Because this area is still evolving, a position paper proposing architectural principles is timely and valuable.

weakness:
1. The writing of this paper needs to be improved, which is redundant and not coherent enough. For example, in the introduction part, the first paragraph on page 2 states that the safety module can be separated into external and internal modularity. However, the third paragraph on page 2 claims that the safety architecture consists of Monitoring and Evaluation Layer and Intervention Layer. The reason of using two kinds of categorization is not clearly stated in the introduction, making readers confused. Besides, in the last paragraph of Section 1, sec. 4 is not introduced. In addition, section 3 can be integrated into section 4. Using too many words on basic concepts is redundant, and the rest of this paper is not built upon these concepts in Section 3.

2. The central claim—using modular safety guardrails to monitor and intervene in AI systems—is not entirely new. Similar ideas already exist in several communities, such as AI alignment / LLM safety [1,4] and robotics safety [2,3]. The LLM safety [1,4] includes guardrails, red-teaming, oversight models, and runtime monitoring. The modular robotic safety guardrails have been investigated in [2,3], including control barrier functions, safety filters, runtime shielding. The paper mainly repackages these ideas into a unified architectural framing, but the conceptual novelty beyond existing “runtime safety layer” paradigms is somewhat limited.

[1] Safety Guardrails for LLM-Enabled Robots, RA-L, 2026
[2] Soter: programming safe robotics system using runtime assurance, arXiv, 2018
[3] SMOF: A safety monitoring framework for autonomous systems, IEEE T-SMC, 2016
[4] Plug in the safety chip: Enforcing constraints for llm-driven robot agents, ICRA, 2024

3. Authors do not argue the major claim of this paper convincingly enough. A strong position paper usually argues a clear, debatable claim with strong reasoning. Here the claim that “modular safety guardrails are necessary” could be seen as too broad or insufficiently justified. The paper asserts necessity but does not rigorously argue why alternative architectures cannot succeed. Especially, the previous modular work on robotic safety is not discussed thoroughly enough. The argument sometimes reads as plausible intuition rather than a logically tight position.

4. The paper attempts to address multiple issues simultaneously, including foundation models in robotics safety (action, decision, and human-centered safety), adversarial attacks, human-robot interaction, and architectural design principles. It is comprehensive. But none of a single aspect is analyzed deeply, making readers feel that the claim proposed by this paper is only high-level and not reasoned deeply.

5. The proposed guardrail architecture introduces additional monitoring and intervention layers, which may significantly increase runtime latency and computational cost. The paper acknowledges this issue but does not provide concrete strategies or analysis to ensure real-time performance for robotic systems.

**Support:**

2

---

> ### Author Rebuttal · Authors · 2026-03-31
>
> **Response to Reviewer AW59**:
>
> We appreciate the reviewer's feedback and address each concern as follows:
>
> **Internal/External Modularity & Two-layer Guardrail Relationship (W1, Q1)**
>
> We will revise the structure of the paper per the reviewer's suggestion, including merging Sec. 3 & 4. We also want to clarify that internal/external modularity & two-layer guardrail are two different levels of description, not two competing categorizations. External modularity and internal modularity are design principles of the guardrail. The Monitoring and Evaluation Layer and the Intervention Layer are the functional architecture that instantiates those principles. External modularity means the guardrail is operationally independent from the upstream FM stack and acts as a distinct safety authority. Internal modularity means the guardrail is itself decomposed into specialized submodules with different roles and time scales. In the current architecture, that internal decomposition appears as the monitoring layer plus the intervention layer, with the intervention layer further split into a decision gate and an action gate. We will revise Sec. 6.1-2 to make this hierarchy explicit.
>
> **Contributions Go Beyond Prior Modular Safety Approaches and Simple Module Stacking (W2, W4)**
>
> Thank you for this important feedback. Our contribution is not to simply repackage existing safety ideas, but to: (1) identify the safety challenges that arise specifically from FM integration in real-world robot deployment, and (2) propose a principled co-design framework for modular safety guardrails (see our response to **TXRr-W1** for details).
>
> We argue that simple module stacking is insufficient for safety. As discussed in Sec. 5, existing action-level, decision-level, and human-centered approaches are individually incomplete, and directly composing them would still leave unresolved coordination problems. Modules designed independently often operate on different representations, so risk information from one module may not be directly usable by another without losing safety-relevant structure. Moreover, if each module applies its own conservative criterion independently, conservatism can accumulate, leading to overly restrictive or even infeasible behavior that does not reflect the true safety risk. These are precisely the issues our co-design principles address. We introduce **representation alignment** and **conservatism allocation** to ensure that safety-relevant information remains actionable across layers and that enforcement responsibilities are coordinated rather than redundantly stacked. We further support this with three concrete deployment examples.
>
> Beyond co-design, we explicitly characterize the architectural requirements any safety mechanism must satisfy for real-world FM-robot deployment: independently verifiable and updateable, composable across complementary failure modes, and non-bypassable at execution time (Sec. 6.1). We then explain how **external** and **internal modularity** support these assurance properties (Sec. 6.2). These requirements and properties, to our knowledge, have not been systematically articulated in prior modular safety work. We will revise the paper to make this distinction and contribution more explicit.
>
> Finally, as a position paper, it is intentionally structured to provide a unified guardrail across the realized aspects of safety (Sec. 3) for FM-enabled robots, instead of treating them as isolated problems. The depth of our argument lies in how these aspects are systematically connected through the proposed modular framework (Sec 6.2.1-2), and how treating them individually fails to address safety in real-world deployment (Sec. 5). The examples (Sec 6.3 and Appx. A) demonstrate how the monitoring and intervention layers operate across the robotic autonomy stack under different failure modes, and how the proposed co-design principle enables less conservative yet effective safety assurance.
>
> **Justification for the Necessity of Modular Architecture (W3)**
>
> Sec. 5 analyzes why each alternative architecture is insufficient in isolation: model-internal safety (View A) lacks a non-bypassable runtime boundary; action-level add-ons (View B1) cannot address semantic or human-centered hazards; decision-level add-ons (View B2) cannot guarantee hard physical safety; and human-centered add-ons (View B3) cannot enforce hard constraints under misalignment. Under this framework, [2, 3] fall under B1, while [1,4] fall under B2; [1] is already discussed in Sec. 5.2 on limitations of decision-level approaches. Collectively, these insufficiencies motivate the need for modular safety guardrails with the properties discussed in Sec. 6.1, and further ground our co-design principles of the modular architecture, as elaborated in our response to W2. We will revise Secs. 5-6 to make this logic more explicit.
>
> **W5 and Q2**: Please refer to the response to **FGki W2**
> **Q3-5**: Please refer to the response to **TXRr W3-4**.

---

> > ### Author Rebuttal · Reviewer_AW59 · 2026-04-04
> >
> > I will increase my rating by one notch.

---

### Official Review · Reviewer_FGki · 2026-03-13

**Significance:** 3
**Argument Clarity:** 3
**Rating:** 5
**Confidence:** 4

**Questions:**

I would like the authors to please address my comments made in the "weaknesses" section above.

**Alternative Views Section:**

Yes

**Compliance With Llm Reviewing Policy A Conservative:**

Affirmed.

**Discussion Potential:**

3

**Paper Summary:**

This paper presents the position that modular safety guardrails are necessary for autonomous robot systems that rely on foundation models for perception, reasoning, and/or control. The emergence of LLM, VLM, and VLA backbones trained on large datasets is rapidly reshaping the robotic autonomy stack, and this paper argues that addressing safety in a similar end-to-end fashion (e.g. by baking safety into the foundation models themselves) is insufficient. The paper first provides an overview of how foundation models are currently leveraged across perception, reasoning, and control, and then proceeds to define three distinct types of safety (action safety, decision safety, human-centered safety) that naturally exist at different levels of abstraction within the autonomy stack, as well as discuss the safety challenges with foundation model-enabled robotics. The paper presents alternative views (safety can be baked into the model; safety as external modules but typically focus on a single layer of the autonomous stack) and then proceeds to discuss its proposed modular safety guardrails which address each of the three aforementioned types of safety through a safety stack that consists of a monitoring layer and an intervention layer. The paper concludes with a discussion of the limitations of the proposed modular safety guardrails (it lives on top of foundation models so it can enforce safety but does not make the models inherently safer; the safety module may itself be a foundation model in which case its failures may correlate with that of the module of which it is supposed to enforce safety; and finally also computational overhead at inference time **[Reviewer comment: and system complexity vs. an end-to-end approach to safe autonomy, this is not stated explicitly in the limitations section]**. The paper concludes with a call to action which rather predictably calls for work that either (i) proposes composable safety modules that fit into this modular view of safety, and (ii) system-level co-design of safety guardrails.

**Position:**

Yes

**Position In Title:**

Yes

**Related Work:**

3

**Strengths And Weaknesses:**

This position paper is generally very well written and easy to follow; I enjoyed reading it. My initial assessment of the paper leans towards acceptance. I list identified strengths as weaknesses as follows:

**Strengths:**
- The paper is well written and easy to follow. It is well organized, positions itself clearly wrt. existing literature and current trends within the area of robot learning, and provides sufficient context for readers unfamiliar with the nuances of safety in robotics.
- The arguments in the paper are logically sound and the paper is likely to result in interesting debate. The authors provide sufficient support for their arguments and the discussion is grounded in contemporary robot learning literature. I appreciate the diagram in Figure 2 as it very clearly describes the proposed safety guardrails.
- Alternative views are reasonable and well grounded in recent literature, with plenty of references.

**Weaknesses:**
- I understand that this is a robotics safety paper, but given that perception, LLMs, and VLMs are relatively more mature research areas and these communities have considered safety for quite some time, it would perhaps be useful to draw connections to work in these areas. Perhaps the authors could include a brief discussion of how safety is currently being addressed in these communities, as well as the ways in which safety for robotics requires additional considerations.
- When discussing limitations, I would perhaps also include a comment on system complexity vs. an end-to-end approach to safe autonomy, since this is not stated explicitly.
- The question of modularity vs. end-to-end may potentially also have a time horizon component to it: it seems rather plausible that a modular approach to safety is the best path forward in the near term, but that perhaps an end-to-end approach may win out in the long run as we have seen time and time again in other areas of machine learning; I didn't find any discussion on this so perhaps that would be good to include as well when discussing alternative views.

**Support:**

3

---

> ### Author Rebuttal · Authors · 2026-03-31
>
> **Response to Reviewer FGki:**
>
> We thank the reviewer for the positive assessment and constructive suggestions. We address them below:
>
> **Method Review for Perception/LLM/VLM Safety (W1)**
>
> Thanks for this helpful suggestion. We agree that making the connection with perception, LLM, and VLM  more explicit would strengthen the paper. We will revise the manuscript to more clearly distinguish two points. (1) The first point is that FM-based robotics inherits several concerns already studied in those communities, including hallucination, weak grounding, uncalibrated uncertainty, and multimodal attack surfaces. We will strengthen this point in Sec. 4 where the current paper already discusses FM-related safety challenges that are also central in LLM/VLM settings, as well as in Sec. 5 where we summarize several representative safety methods from these communities (e.g., model-internal alignment, uncertainty quantification, LLM-as-a-Judge, semantic filtering, and preference modeling). (2) The second point is that robotics introduces additional unique challenges, physical constraints, real-world disturbances, open-world deployment, and direct physical interaction with humans, which make failures propagate beyond prediction errors into unsafe physical behavior and therefore require execution-time, non-bypassable safeguards. This second point will be further clarified in Sec. 4 where these aspects are currently discussed.
>
> **System Complexity (W2 & AW59-W5,Q2)**
>
> Thanks for raising this point. We agree that modular safety should be discussed more explicitly through the lens of complexity tradeoff. Compared with a fully end-to-end approach, modular guardrails introduce additional components, interfaces, and coordination requirements, which can increase engineering and maintenance complexity. Our view is instead that it can allow limited computation and adaptation effort to be focused on the safety-critical part of the system, rather than requiring a single end-to-end model to absorb all safety requirements. In practice, our architecture is explicitly multi-timescale instead of assuming all monitoring and intervention occur synchronously or at the same frequency. The decision gate handles slower semantic and social screening, and may run asynchronously or selectively on high-risk situations; the action gate provides the real-time, non-bypassable last line of defense at execution time. This separation is precisely why internal modularity matters: it allows expensive semantic reasoning and fast low-level safety enforcement to be decoupled instead of forcing one monolithic safety component to satisfy incompatible timing requirements. We will make this design rationale explicit and strengthen Sec. 6.4 to clarify that real-time enforceability rests on keeping action-level safety mechanisms lightweight and execution-facing, while higher-level checks can be parallelized or triggered conditionally, and further discuss this complexity tradeoff more explicitly.
>
> **End-to-end in the Long Term (W3)**
>
> Thanks for raising this important question. We agree that discussing this perspective in Alternative Views would strengthen the paper. We also agree that end-to-end safety may continue to improve over time. However, our view is that such progress is more likely to reduce reliance on modular safeguards in controlled lab settings than to eliminate their value in open-world, safety-critical deployment. In our view, stronger end-to-end models and external modular safeguards are not competing choices. Rather, they can work together: improved end-to-end models may enhance average-case behavior, while modular guardrails provide an independent, non-bypassable, and externally auditable layer of protection, especially for rare failures, evolving environments, and shifting safety requirements. We will add this as an Alternative View and clarify that our claim is strongest for real-world deployment beyond controlled lab settings.
>
> Relatedly, emerging results in end-to-end VLA systems also suggest that modular adaptation can remain useful even within otherwise end-to-end pipelines. For example, recent work such as RL Token from Physical Intelligence [1] explores adapting behaviors through a task-specific residual module rather than fine-tuning the entire VLA, partly for computational efficiency. By analogy, we believe achieving universally reliable safety across all applications with a single monolithic end-to-end model will be difficult, while fully fine-tuning a large VLA to meet the safety requirements of every downstream domain may be prohibitively costly. In contrast, adapting a modular safety guardrail to the requirements of a specific deployment domain may often be a more feasible and effective path. We will incorporate these discussions in Sec.5.1 to broaden the discussion on Alternative Views.
>
> [1] Charles Xu, et al. "RL Token: Bootstrapping Online RL with Vision-Language-Action Models " https://www.pi.website/research/rlt

---

> > ### Author Rebuttal · Reviewer_FGki · 2026-04-03
> >
> > Thank you for the detailed response! I was already leaning towards acceptance and my concerns can all be addressed for the camera-ready version which the authors have committed to, so I will keep my score.

---

### Official Review · Reviewer_CQxd · 2026-03-13

**Significance:** 4
**Argument Clarity:** 3
**Rating:** 5
**Confidence:** 2

**Questions:**

Are all safety properties in embodied AI monitorable? Any thought on the decidability/complexity issues that relate to your position?

**Alternative Views Section:**

Yes

**Compliance With Llm Reviewing Policy A Conservative:**

Affirmed.

**Discussion Potential:**

4

**Final Justification:**

The rebuttal is satisfactory, However, my recommendation of accept adequately reflects the current state of the paper.

**Paper Summary:**

The paper presents the position that architectural modularity is essential for the safe deployment of embodied foundation models. This is a timely topic with great potential interest in the near future.

**Position:**

Yes

**Position In Title:**

Yes

**Related Work:**

3

**Strengths And Weaknesses:**

Strengths:
1. The paper captures the role of foundation models as perception, reasoning and action models, and discusses the safety aspects of each of these.
2. The position paper studies challenges to safety from distribution shifts, long-tail failures, adversarial inputs, open-world co-deployment with human presence.
3. The architecture with monitoring & evaluation as well as decision, action, and safety assurance is clearly articulated.

Weaknesses:
1. The example of the LLM planner discussed in the paper does not adequately support all aspects of the architecture in the position paper.
2. The position paper does not discuss how it is informed by complexity-theoretic and decidability arguments.

**Support:**

3

---

> ### Author Rebuttal · Authors · 2026-03-31
>
> **Response to Reviewer CQxd:**
>
> We thank the reviewer for the positive assessment of our paper. We also appreciate the constructive suggestions raised here, we will address them in detail as follow:
>
>
>
> **Response to W1: LLM Planner Example**
>
> Thank you for raising this point. We agree that a single LLM planner example cannot fully capture all aspects of the broader architecture discussed in the position paper. To address this within the page limit, we included two additional examples in the appendix: one centered on foundation models used for perception, and another on end-to-end VLAs, illustrating mechanisms of monitoring layer, action gate, decision gate, and the co-design principles. We would also welcome suggestions on any specific architectural setting that remains under-illustrated, and we would be happy to elaborate further and include additional examples.
>
>
>
> **Response to W2 and Q2: Complexity-theoretic and Decidability Arguments**
>
> Thank you for this thoughtful comment. We agree that the paper should more clearly explain how the complexity-theoretic and decidability issues are related to our position. Our understanding is that decidability concerns whether a safety property can, in principle, always be formally determined by an algorithm, while complexity concerns whether such verification is computationally tractable, especially under real-time constraints. While this position paper is not primarily centered on complexity-theoretic or decidability-based analysis, we agree that this perspective would provide a useful complement to our discussion. One motivation for modularity in our framework is precisely to allow different mechanisms to be designed and optimized according to their own computational complexity requirements, with varying levels of formal specification, without compromising the overall system's real-time operation, as discussed in Sec. 6.4. We will add a short discussion in the paper to clarify this connection and better position modularity as a response to heterogeneous complexity requirements and decidability.
>
>
>
> **Response to Q1: Monitorability**
>
> Thank you for raising this important point. We agree that the paper should more clearly explain how formal limits on monitoring relate to our position. We would like to clarify that we do not assume that all safety properties in embodied AI are fully decidable or efficiently monitorable. Low-level physical safety properties such as joint limits, collision margins, or velocity bounds are often amenable to runtime monitoring or enforcement. In contrast, higher-level semantic or human-centered safety properties are typically only partially specifiable and may require approximate, probabilistic, or human-in-the-loop assessment. Even when a safety metric is decidable in principle, exact online checking may still be too computationally expensive for real-time embodied deployment. Our position is therefore not that modularity resolves these formal issues in general, but that it helps decompose the safety problem into subproblems with clearer specifications and more tractable monitoring or intervention mechanisms. We will add a paragraph in Sec. 6.2.1 clarifying the distinction between runtime-monitorable physical constraints and higher-level partially specifiable safety properties.

---

> > ### Author Rebuttal · Reviewer_CQxd · 2026-04-02
> >
> > The rebuttal is satisfactory, However, my recommendation of accept adequately reflects the current state of the paper.

---

### Decision · Program_Chairs · 2026-04-30

**Decision:**

Accept (regular)

**Comment:**

The position paper argues that existing approaches, such as static verification or end-to-end learned policies, are insufficient for open-ended, real-world settings and proposes modular safety guardrails (consisting of monitoring and intervention layers) as an architectural foundation for comprehensive robot safety. In essence, the position makes the case that safety in foundation-model-based robotics must be treated as a separate, modular concern rather than something baked into the model itself. Overall, the reviews lean towards acceptance (3x accept, 1x borderline reject). The main downside raised is a potentially weak novelty, as using modular safety guardrails to monitor and intervene in AI systems is not entirely new. As pointed out by one reviewer, similar ideas already exist in several communities, such as AI alignment / LLM safety and robotics safety. While indeed the claim that “modular safety guardrails are necessary” could be seen as too broad or insufficiently justified, the overall takeaway message that modularity is important is still a very strong one with also connections to hybrid and neuro-symbolic AI. So overall this paper should be accepted.